# Implications of Podoplanin Overexpression in the Malignant Transformation of Oral Potentially Malignant Disorders: A Systematic Review and Meta-Analysis

**DOI:** 10.3390/cancers17213448

**Published:** 2025-10-28

**Authors:** Marcela Correa-Fernández, Pablo Ramos-García, Noor Mjouel-Boutaleb, Hajar Boujemaoui-Boulaghmoudi, Miguel Ángel González-Moles

**Affiliations:** 1Facultad Odontología, Universidad Mayor Chile, Temuco 4810296, Chile; marcecorrea2@yahoo.es; 2School of Dentistry, University of Granada, 18011 Granada, Spain; noormb@correo.ugr.es (N.M.-B.); hajarbb@correo.ugr.es (H.B.-B.); 3Instituto de Investigación Biosanitaria ibs.GRANADA, 18012 Granada, Spain

**Keywords:** podoplanin, pdpn, oral potentially malignant disorders, oral leukoplakia, oral erythroplakia, oral lichen planus, oral cancer, systematic review, meta-analysis

## Abstract

**Simple Summary:**

Oral cancer represents a global health problem, with an estimated annual incidence of 377,713 new cases and 177,757 deaths. Despite advances in oral oncology, its prognosis has remained largely unchanged over the past 40 years, with a 5-year survival rate close to 50%. Oral cancer is often preceded by oral potentially malignant disorders (OPMDs), which are defined by the WHO Collaborating Centre for Oral Cancer as mucosal alterations associated with an elevated risk of progression to oral cancer. At present, no reliable method exists to accurately determine which individuals with OPMDs will undergo malignant progression. Consequently, ongoing research is exploring molecular biomarkers as potential predictive tools for assessing the risk of OPMDs malignant transformation. Among these biomarkers, podoplanin emerges as a promising biomarker with potential predictive value of malignant transformation in oral leukoplakias. Despite the importance of this topic, it is noteworthy that, to date, there is no evidence, in the form of systematic reviews and meta-analyses, on the implications of podoplanin in the malignant transformation of OPMDs.

**Abstract:**

Objective: To evaluate the degree of current evidence through a systematic review and meta-analysis on the association between podoplanin overexpression and the malignant transformation of oral potentially malignant disorders (OPMDs). Methods: A systematic search was performed in the MEDLINE (through PubMed), Embase, Scopus, and Web of Science databases for primary-level research published before December 2024, strictly designed as longitudinal cohorts with follow up data, and no restrictions by language or publication date. The Quality in Prognosis Studies QUIPS tool (developed by the Cochrane Prognosis Methods Group) was applied in order to assess the methodological quality and risk of bias. Meta-analyses, subgroup meta-analyses, sensitivity, and small-study effects analyses were performed. Results: Twelve primary-level studies met the eligibility criteria and were included, which followed up 857 OPMDs patients over time. Podoplanin overexpression was significantly associated with an increased risk of the malignant transformation of OPMDs (RR = 3.64, 95% CI = 2.18–6.10, *p* < 0.001). Podoplanin also proved to be a valuable biomarker in the malignant transformation of all investigated OPMDs (oral leukoplakia: *p* < 0.001; erythroplakia: *p* = 0.05; oral lichen planus: *p* = 0.02; discoid lupus erythematosus: *p* = 0.009). In addition, podoplanin overexpression was significantly associated with an increased risk of cancer development in several study subgroups with methodological implications (anti-podoplanin D2-40 antibody: *p* < 0.001; membrane and cytoplasm staining: *p* < 0.001; antibody dilution at 1:100: *p* < 0.001; overnight incubation: *p* < 0.001; 4 °C incubation: *p* < 0.001; cut-off point > 1%: *p* < 0.001; low risk of bias: *p* < 0.001). Conclusions: This systematic review and meta-analysis, on the basis of the evidence, indicates that podoplanin overexpression is a predictor of an increased risk of malignant transformation in OPMDs, singularly in oral leukoplakias. Its evaluation using immunohistochemical methods would be advisable in pathology laboratories.

## 1. Introduction

In March 2020, the WHO Collaborating Centre for Oral Cancer convened a workshop meeting of international experts to provide updated knowledge on the nomenclature, classification, and malignant transformation of oral potentially malignant disorders (OPMDs). These disorders encompass a spectrum of lesions and conditions of the oral mucosa that are associated with an elevated risk of progression to oral cancer [1]. Oral squamous cell carcinoma (OSCC) constitutes the most prevalent subtype of oral cancer, comprising nearly 90% of all malignant neoplasms arising in the oral cavity [2]. Globally, oral cancer is estimated to affect approximately 377,713 individuals annually, resulting in around 177,757 deaths each year (GLOBOCAN, IARC, WHO) [3], and a 5-year mortality rate approaching 50% that has not changed in the last 40 years [2]. Oral cancer’s poor prognosis is mainly due to diagnostic delay [4], which constitutes a complex polyhedral problem that is difficult to resolve due to the numerous factors involved (related to the patient, to the professionals, and to health policies, among others). In this sense, the clinical management and follow-up of OPMD patients is complex, and currently there are no efficient tools that allow us to accurately predict which cases will evolve into OSCC.

In recent decades, research has been carried out into the molecular alterations underlying OPMDs, which may be regarded as early events in oral carcinogenesis and, therefore, could be used as predictive biomarkers of malignant transformation. In this regard, the protein podoplanin has received considerable interest, mainly due to the relevant findings reported by a recent consensus meeting of the WHO Collaborating Centre for Oral Cancer [5], and the international research initiative of the World Workshop on Oral Medicine [6]. Both investigations concluded that podoplanin is one of the most promising and predictive biomarkers of the malignant transformation of oral leukoplakias [5,6]. Podoplanin is a transmembrane glycoprotein with oncogenic roles that appear to participate in the regulation of the epithelial–mesenchymal transition (EMT) phenomenon. This is a well-established molecular phenomenon in which cells of epithelial lineage—with polygonal or epithelioid morphology—acquire a spindle-shaped or fibroblastoid structural conformation due to dynamic changes in the actin cytoskeleton [7]. These molecular and cellular events allow the cell to enhance cell motility and acquire a migratory phenotype and a canonical cancer hallmark (i.e., activating invasion and metastasis) [8,9], with the consequent gain of oncogenic advantages that allow epithelial cells to invade the underlying tissues.

Despite the importance of this topic, evidence from systematic reviews and meta-analyses on the implications of podoplanin in the malignant transformation of OPMDs remains limited and derives mainly from oral leukoplakias [10,11]. Therefore, we developed the present systematic review and meta-analysis with the aim of qualitatively and quantitatively evaluating the current evidence on the predictive value of podoplanin in the malignant transformation of OPMDs. High-standard methodological criteria were followed and only primary-level studies that strictly followed patients over time were analyzed, allowing causal relationships between podoplanin overexpression and oral cancer development to be established. Furthermore, we point out the most plausible clinical applications of our findings.

## 2. Materials and Methods

This systematic review with meta-analysis was carried out in accordance with the PRISMA (Preferred Reporting Items for Systematic Reviews and Meta-Analyses) statement and the MOOSE (Meta-analysis Of Observational Studies in Epidemiology) guidelines for structured reporting [12], as well as the methodological principles outlined by the Cochrane Prognosis Methods Group and the Cochrane Handbook for Systematic Reviews of Interventions [13,14].

### 2.1. Protocol

The methodology for the study was developed in the previous phase and documented in the international prospective register of systematic reviews PROSPERO (registration number: ID1081708/CRD420251081708). This approach was adopted to ensure transparency, accuracy, and integrity, thereby reducing potential bias. The original full version of the protocol was developed in accordance with the PRISMA-P standards to ensure methodological accuracy [15].

### 2.2. Search Strategy

A systematic review was conducted in four databases: MEDLINE (via PubMed), Embase, Web of Science, and Scopus. The purpose was to identify all relevant studies published before the upper date limit of December 2024, including studies published at any time prior to the search date. A combination of thesaurus terms (Emtree and MeSH) and free terms was used to increase the sensitivity of the search strategy (the full syntax can be found in Appendix A). Furthermore, the lists of references of initially selected primary-level studies were carefully checked. References were managed and duplicates removed using Mendeley, version 1.19.8 (Elsevier, Amsterdam, The Netherlands).

### 2.3. Eligibility Criteria

Studies were required to fulfill the following criteria for inclusion in this systematic review: (1) any language and year of publication; (2) the study population included patients with any oral potentially malignant disorder (OPMD); (3) podoplanin upregulation analyzed by immunohistochemical protein overexpression; (4) data on the risk of malignant transformation (progressor and non-progressor cases) to OSCC; and (5) prospective or retrospective study cohorts of a longitudinal nature.

The exclusion criteria were as follows: (1) reviews, meta-analyses, case reports, editorials, letters, retracted articles, book chapters, abstracts of scientific meetings, comments, and personal opinions; (2) studies conducted in in vitro or in vivo animal models; (3) research on podoplanin genetic alterations modifications different to protein overexpression; (4) OSCC studies lacking statistical data on malignant transformation or with insufficient statistical data to calculate relative risk (RR) with 95% confidence interval (CI); (5) cross-sectional or interventionist study designs; and (6) articles that involve the same population, as identified by crosschecking authorship and affiliation, confirming patient source, and examining recruitment periods.

The study selection process was carried out in two phases; to identify primary-level studies that met the inclusion criteria, we first screened the titles and abstracts of all studies. Later, the full texts were then reviewed for compliance against the precedent exclusion criteria.

### 2.4. Data Extraction

The authors independently extracted data from the sampled articles. They completed a standardized data collection form using Excel (version 16/2018, Microsoft, Redmond, WA, USA). The initial presentation of the data, in the form of order statistics (i.e., median and/or maximum–minimum values) were converted into means ± standard deviation (SD) according to the methods proposed by Luo et al. (2018) and Wan et al. (2014) [16,17]. In cases where it was necessary to combine two or more datasets, expressed as means ± SD from subgroups into a single group, the Cochrane Handbook formula was used [18]. The datasets extracted included the following information: name of the first author, year of publication, the continent and country, the language, the study design, the recruitment and follow-up period, the sample size, the anatomical site, and demographics such as age and sex of patients, tobacco and alcohol use, type of OPMD, details of oral epithelial dysplasia, and progression or non-progression to oral cancer. All studies assessed the expression of this protein using the immunohistochemical technique. Laboratory variables (i.e., type of antibody, dilution, incubation time and temperature, type of immunohistochemical pattern), the main methods of quantification, and the scoring system, such as cut-off point and percentage of podoplanin overexpression, were also collected.

### 2.5. Evaluation and Risk of Bias

The Quality in Prognosis Studies (QUIPS) tool was used to critically appraise the methodological quality and risk of potential bias across the included studies, as recommended by the Cochrane Prognosis Methods Group [19]. The risk of bias assessment was implemented across standardized domains commonly associated with a high potential for bias in systematic reviews addressing prognostic factors, which are specifically referred to as follows: (1) study participation, (2) study attrition, (3) prognostic factor measurement, (4) outcome measurement, (5) study confounding, and (6) statistical analysis and reporting. Each domain was evaluated according to a three-level scale reflecting a low, moderate, or high risk of bias. An overall score was also computed by our research group using a previously established methodology based on the two critical areas (No. 3 and No. 5) [20,21,22,23,24]. This was carried out with the aim of statistically assessing the impact of primary study methodological quality on our meta-analysis results.

### 2.6. Statistical Analysis

Podoplanin protein overexpression was analyzed as a dichotomous categorical variable in accordance with the scoring systems adopted by primary-level studies. The relative risks (RRs) with 95% confidence intervals (CIs) were calculated from the studies at the primary level and then pooled using the method of inverse variance in a random-effects model based on the DerSimonian and Laird classical method [25], to account for potential variability between study subpopulations (e.g., differences between OPMDs, podoplanin laboratory detection techniques, and affected oral subsites).

Statistical heterogeneity was assessed using the Cochran’s Q test. In light of the limited statistical power of this test, a significance level of *p* < 0.10 was considered significant. To further quantify the heterogeneity among studies, Higgins’ I^2^ statistic was calculated. This index estimates the percentage of total variability in effect estimates that is due to true differences between studies rather than random sampling error. Values ranging from 50% to 75% were considered indicative of a moderate-to-high level of inconsistency across studies [26,27]. Planned subgroup meta-analysis series were conducted to identify potential sources of heterogeneity and to investigate the most important factors associated with podoplanin overexpression and OPMD malignant transformation [28]. To provide a visual representation of the overall effects, forest plots were constructed for each meta-analysis in order to facilitate sequential visual inspection analyses.

Furthermore, secondary analyses were also conducted to assess the stability and reliability of the meta-analytical results. To evaluate the influence of the individual studies on the pooled estimate, sensitivity analyses were performed [29]. This was achieved using a leave-one-out approach, in which the meta-analysis was repeated sequentially, with each study being omitted in turn. Finally, small-study effects were analyzed to identify potential biases—such as publication bias—by conducting funnel plots, the Egger regression test (*p*Egger < 0.10 was considered as significant asymmetry) and the non-parametric trim-and-fill technique [30,31]. All statistical analyses were performed using Stata software (version 16.1, Stata Corp, College Station, TX, USA).

## 3. Results

### 3.1. Results of the Literature Search

Figure 1 illustrates the process employed for the identification and selection of studies in this systematic review. Initially, 837 records were retrieved from electronic databases: 113 from Web of Science, 342 from Embase, 283 from Scopus, and 99 from MEDLINE/PubMed. After removing duplicates, 555 unique records remained and were screened based on title and abstract. From these, 28 articles were deemed potentially relevant and assessed in full text. Of those, 16 were excluded for not fulfilling the predefined eligibility criteria. Detailed reasons for exclusion are provided in the Appendix A. Consequently, 12 primary-level studies met all inclusion criteria and were incorporated into the final qualitative and quantitative evaluation [32,33,34,35,36,37,38,39,40,41,42,43].

### 3.2. Study Characteristics

Table 1 summarizes the main characteristics of the selected studies, while the extended characteristics of each primary study are displayed in Appendix A. A total of 857 patients with OPMD were recruited across 12 studies published between 2008 and 2022. According to the eligibility criteria, all studies provided data on malignant transformation across follow up (*n* = 12). The number of participants in each study ranged from 30 to 160. The studies were performed in Asia (*n* = 8), Europe (*n* = 3), and North America (*n* = 1). Ten were retrospective in design, while the remaining two were prospective. Oral leukoplakia was the OPMD most frequently investigated (*n* = 7). All studies evaluated podoplanin expression via immunohistochemistry (*n* = 12); most studies used the anti-podoplanin monoclonal antibody D2-40 (*n* = 10), incubated overnight (*n* = 5) at 4 °C (*n* = 5) with a dilution of 1:100 (*n* = 6) and used a cut-off point ≥1 for podoplanin overexpression (*n* = 4).

### 3.3. Qualitative Evaluation

The qualitative analysis was performed with the aid of the QUIPS (Quality In Prognosis Studies) tool, which assesses the potential for bias in six domains (Figure 2).

*Study participation*: A high potential risk of bias was present in 25.00% of the studies, while 33.33% harbored a moderate risk and 41.67% low risk. The most frequently identified sources of bias were the inadequate description of the patients’ clinicodemographic characteristics (e.g., sex, age, or lesion locations), lack of information regarding the recruitment setting, and study period.

*Study attrition;* A high risk of bias was identified in 16.66% of cases, a moderate risk in 41.67%, and a low risk in 41.67%. The most frequent biases were related to the omission of crucial information regarding dropout rates or the duration of follow-up periods.

*Prognostic factor measurement*: The risk of bias in this domain was high in 50.00% of the studies, moderate in 41.67%, and low in 8.33%. The most relevant source of bias was the use of optimized cut-off point design and analysis, along with the lack of reporting on this essential methodological aspect for evaluating a prognostic factor. Furthermore, some studies failed to report essential details about the laboratory methods used for immunohistochemical techniques, making subsequent replication difficult.

*Outcome measurement*: In this domain, the risk of bias was identified as moderate in 58.33% and low in 41.67% of the reviewed studies, while no study was critically judged as having a high risk of bias. This domain achieved the highest ratings, given that the clinical and histopathological approaches used to confirm OSCC development are widely recognized as standard practice. Moreover, the absence of explicit details regarding the diagnostic system was not interpreted as a source of potential bias.

*Study confounding*: A high risk of bias was observed in 33.33% of the studies, moderate in 16.67%, and low in 50.00%. Some studies did not adequately design or analyze the presence of potential confounding factors.

*Statistical analysis and reporting*: Few studies (8.33%) showed a high risk of bias, 33.33% a moderate risk, and 58.33% a low potential risk of bias. In this domain, the most frequent bias was the omission by some studies of essential statistical metrics needed to evaluate the magnitude, direction, and precision of the effect investigated (i.e., RRs and 95% CIs).

### 3.4. Quantitative Evaluation (Meta-Analysis)

*Meta-analysis on the association between podoplanin overexpression and the malignant transformation risk of OPMDs:* The random-effects model indicated a significantly increased risk among patients with OPMDs and podoplanin expression (RR = 3.64; 95% CI = 2.18–6.10; *p* < 0.001). In addition, low statistical heterogeneity was observed across studies (*p* = 0.10, I^2^ = 37.1%) (Figure 3, Table 2).

*Subgroups meta-analysis*: A significant association between podoplanin expression and malignant transformation in OPMDs was observed across several subgroups, stratified by geographical area (Asia: RR = 3.20, 95% CI = 1.61–6.34, *p* = 0.001; Europe: RR = 9.66, 95% CI = 3.19–29.26, *p* < 0.001; North America: RR = 2.81, 95% CI = 1.40–5.66, *p* = 0.004), by OPMD (oral leukoplakia: RR = 3.37, 95% CI = 2.13–5.32, *p* < 0.001; erythroplakia: RR = 6.31, 95% CI = 1.02–38.98, *p* = 0.05; discoid lupus erythematosus: RR = 18.67, 95% CI = 2.07–168.82, *p* = 0.009; oral lichen planus: RR = 17.13, 95% CI = 1.71–171.4, *p* = 0.02), by immunohistochemical pattern (membrane and cytoplasm: RR = 4.71, 95% CI = 2.54–8.74, *p* < 0.001; membrane: RR = 3.60, 95% CI = 1.96–6.62, *p* < 0.001), by anti-podoplanin antibody (clone D2-40: RR = 3.45, 95% CI = 1.95–6.10, *p* < 0.001), by antibody dilution (1:100: RR = 2.62, 95% CI = 1.51–4.56, *p* = 0.001; 1:150: RR = 11.04, 95% CI = 4.00–30.45, *p* < 0.001), by incubation time (overnight: RR = 3.82, 95% CI = 2.11–6.92, *p* < 0.001; 1 h: RR = 4.67, 95% CI = 0.67–32.56, *p* = 0.12), by incubation temperature (4 °C: RR = 3.82, 95% CI = 2.11–6.92, *p* < 0.001; room temperature: RR = 4.67, 95% CI = 0.67–32.56, *p* = 0.12), by cut-off point (1%: RR = 3.42, 95% CI = 1.77–6.63, *p* < 0.001; intensity-based: RR = 2.85, 95% CI = 0.35–22.90, *p* = 0.33), and by overall risk of bias (low: RR = 5.20, 95% CI = 2.16–12.51, *p* < 0.001; moderate: RR = 3.70, 95% CI = 2.01–6.82, *p* < 0.001) (Appendix A, Table 2).

### 3.5. Quantitative Secondary Analyses

*Analysis of small-study effects:* The asymmetry of the funnel plot and the results of the corresponding statistical test (pEgger = 0.009) pointed out the presence of small-study effects on the malignant transformation variable, suggesting publication bias. This was confirmed after the application of the non-parametric trim-and-fill method, which identified five missing studies (Appendix A). To assess the magnitude of publication bias and provide an overall adjusted estimate, we re-conducted the meta-analysis incorporating these imputed studies using the same statistical model. The adjusted pooled effect remained statistically significant (*p* = 0.002) and continued to show a large effect size (RR = 2.39, 95% CI = 1.38–4.13), reinforcing the robustness and reliability of our findings despite the potential influence of publication bias.

*Sensitivity analysis:* The overall findings remained consistent throughout the sequential “leave-one-out” sensitivity analyses, in which each study was omitted individually in turn (Appendix A). This indicates that the estimated pooled relative risks are not disproportionately influenced by any single primary study, thereby supporting the stability and robustness of meta-analytical results.

## 4. Discussion

Our systematic review and meta-analysis of the implications of podoplanin in the malignant transformation of OPMDs, carried out across 12 primary-level studies including 857 OPMD patients, demonstrates that podoplanin overexpression is significantly associated with an increased risk of OSCC development in these patients (RR = 3.64, 95% CI = 2.18–6.10, *p* < 0.001). This is important because we do not currently have a predictive tool that accurately discriminates which OPMDs will undergo malignant transformation over time. This aspect is very relevant for the clinical management of OPMD patients and for the prognosis of the disease because OSCC is an aggressive and rapidly progressing neoplasm, and early diagnosis is considered the only feasible strategy to improve the prognosis of patients [4]. In this sense, the immunohistochemical assessment of podoplanin could be useful as a predictive marker for the malignant transformation of OPMD, opening a window of opportunity in those patients showing protein overexpression. Podoplanin overexpression has also been associated with poorer prognosis in oral and oropharyngeal squamous cell carcinomas, including reduced overall survival, disease-specific survival, and disease-free survival, as well as a higher frequency of nodal metastasis (N+ status) [44]. Similar associations have been reported in other malignancies, most notably in lung, esophageal, and ovarian cancers [45,46], where podoplanin expression correlates with unfavorable clinical outcomes. This behavior appears to be linked to the regulation of epithelial–mesenchymal transition (EMT) mediated by podoplanin [47,48], which confers tumor cells with a pro-migratory phenotype and provides an oncogenic advantage through the acquisition of the cancer hallmark “activating invasion and metastasis” [8,9]. This hallmark of cancer represents a critical step in tumor progression and remains one of the main determinants of cancer-related mortality [8,9,47,48].

Furthermore, this meta-analysis provides specific results on the implications of podoplanin overexpression on the malignant development of the OPMDs investigated. Our results demonstrate that all OPMDs had statistically significant results for podoplanin overexpression and malignant transformation. Oral leukoplakia was the most investigated lesion, and for which there is the most evidence for the predictive value of podoplanin overexpression, which was significantly associated with an increased malignant transformation risk (RR = 3.37, 95% CI = 2.13–5.32, *p* < 0.001; 7 studies/558 patients). Oral leukoplakia, given its considerable prevalence (ranging from 1.36% to 4.11% [49,50,51]) and its elevated malignancy rate (6.64% [52]) makes it a clinically relevant OPMD. A recent meta-analysis published by our research group also pointed out that the risk of developing cancer is singularly higher for non-homogeneous leukoplakia (RR = 4.23) of larger clinical size (RR = 2.08) located in the margin of the tongue (RR = 2.09) in patients with a smoking habit (RR = 1.64) and with the presence of oral epithelial dysplasia (RR = 2.75) [52]. In another meta-analysis conducted by our research group evaluating the molecular markers of malignant transformation in oral leukoplakia, we reported an increased risk of oral cancer development in those leukoplakias that acquire a sustained proliferative capacity, along with the ability to evade antitumor immune responses and to activate mechanisms related to increased invasiveness and metastatic potential [53]. The latter includes the actions of podoplanin, which, as previously discussed, is a protein involved in EMT processes. We can now also state on the basis of the evidence that oral leukoplakias that overexpress podoplanin have a risk of malignancy more than three times higher than those that do not overexpress this protein. Consequently, in addition to the other clinicopathological parameters discussed above, which have proven potential utility in assessing the malignant transformation risk of oral leukoplakia, the analysis of immunohistochemical overexpression of podoplanin should also be advisable.

The currently available evidence is lower for other OPMDs investigated, with all of them respectively analyzed by a single study, but the results obtained were promising in all cases. Podoplanin overexpression was significantly associated with an increased risk of malignant transformation of erythroplakia (RR = 6.31, 95% CI = 1.02–38.98, *p* = 0.05). Erythroplakia is a rare lesion, and although there is insufficient evidence on its actual malignancy rate [54], its very high risk of malignant transformation is well recognized [55], which justifies the importance of its early diagnosis. Another interesting result was obtained for oral lichen planus, where podoplanin overexpression was significantly associated with a very increased risk of developing oral cancer (RR = 17.13, 95% CI = 1.71–171.4, *p* = 0.02). It should be noted that in health sciences, a very large effect size is now considered when a metric ratio is higher than 5, and it involves a considerable upgrade in the assessment of the quality of evidence according to influential international consensus guidelines [56,57]. Therefore, the relative risk obtained in our meta-analysis, higher than 17, should really be taken into consideration and shines a light on a truly promising finding. This reflection forces us to recommend the development of future primary-level studies to confirm, with larger sample sizes, whether podoplanin constitutes a valuable predictive tool for malignant transformation in oral lichen planus patients. Oral lichen planus affects 1% of the general population with a significantly higher prevalence from 40 years [58]. Recently published meta-analyses by our research group show that between 1.14% and 2.28% of cases will develop oral cancer [59,60], making disease awareness and the early diagnosis of oral cancer in these patients imperative. In summary, the present meta-analysis also indicates that the predictive value of podoplanin overexpression in malignant transformation OPMDs is not exclusively limited to oral leukoplakias and represents an important line of future research for other OPMDs, such as erythroplakia or oral lichen planus. A systematic review and meta-analysis published during the final preparation of the present manuscript also examined the association between podoplanin expression and the risk of malignant transformation in OPMDs [11]. Their pooled sample partially overlapped with ours, and the meta-analytic findings were in line with our results, confirming that podoplanin overexpression was significantly associated with an increased risk of oral cancer development, with progressively higher podoplanin expression levels across the disease spectrum (i.e., normal oral mucosa < OPMD < OSCC) [11]. The study was well designed and conducted; however, one of the key aspects of the shared objective—i.e., to evaluate malignant transformation according to specific OPMD subtypes—was not addressed in their analysis [11]. In this regard, the present study represents the first systematic review and meta-analysis specifically designed to explore this dimension, including specific stratified meta-analytic results for oral leukoplakia, oral lichen planus, and oral erythroplakia, providing original and scientifically valuable insights that justify its publication.

The present meta-analysis also allows us to recommend more efficient laboratory methods and systems for quantifying podoplanin expression, derived from results that showed higher performance in analyses stratified by subgroups. In this regard, the use of the Clone D2-40 antibody was widespread, and its recommendation is essential, which is reinforced by the results obtained (RR = 3.45, 95% CI = 1.95–6.19, *p* < 0.001). Our results also allow the recommendation of the cellular compartment in which podoplanin expression should be investigated across combined membrane and cytoplasm expression (RR = 4.71, 95% CI = 2.54–8.74, *p* < 0.001) or via membranous expression, where this protein is usually expressed (RR = 3.60, 95% CI = 1.96–6.62, *p* < 0.001). In addition, as is normally the case, the highest performance was obtained by anti-podoplanin antibody overnight incubation (RR = 3.82, 95% CI = 2.11–6.92, *p* < 0.001) at a temperature of 4 °C (RR = 3.82, 95% CI = 2.11–6.92, *p* < 0.001). Finally, our results recommend the immunohistochemical evaluation of podoplanin overexpression by applying a quantitative count, using a cut-off point ≥ 1% (RR = 3.42, 95% CI = 1.77–6.63, *p* < 0.001). On the contrary, our results advise against the implementation of semi-quantitative methods based on the analysis of expression through labeling intensity, which lost statistical significance in the subgroup meta-analysis (*p* = 0.33). Most research groups quantified podoplanin overexpression using a 1% cut-off point or minor adjustments, as podoplanin expression in the oral epithelium is considered an abnormal feature [61]. The most commonly used cut-off defined a sample as positive when podoplanin expression was observed in at least one area of the suprabasal layer (also defined as >1% suprabasal expression), with or without basal expression, based on the rationale that this may reflect clonal expansion and a higher risk of malignant progression [40]. Regarding studies that used staining intensity as a criterion, the methodology involved scoring samples according to staining intensity alone (“none”, “weak”, “moderate”, or “strong”) [32], or in combination with the percentage of stained cells [42]. Taken together, these findings support the incorporation of immunohistochemical assessment of podoplanin overexpression into routine clinical practice for patients with OPMDs, following the recommendations derived from this systematic review and meta-analysis, including the choice of anti-podoplanin clone, the immunostaining pattern, and the antibody’s incubation time and temperature. Immunohistochemistry is a simple, inexpensive, and widely available technique in pathology laboratories worldwide, and podoplanin could serve as a complementary prognostic tool, facilitating risk stratification for malignant transformation and potentially aiding in the early detection of oral cancer.

Among the potential limitations that warrant consideration, it should be noted that several of the included studies lacked the comprehensive reporting of key demographic and clinicopathological variables, such as sex, age, tobacco and alcohol use, and epithelial dysplasia. Nonetheless, these shortcomings reflect intrinsic limitations of the original studies themselves, and as recommended in our risk of bias analysis using QUIPS tool, it highlights the need for greater methodological rigor and standardization in data reporting at the primary-research level. In contrast, heterogeneity—commonly a major issue in systematic reviews and meta-analyses—did not emerge as a significant concern in the present investigation. No substantial clinical, methodological, or statistical heterogeneity was identified, and the findings demonstrated consistency. On the other hand, we identified a significant impact of small-study effects and the presence of missing studies with the non-parametric trim and fill technique, suggesting the presence of publication bias. However, a subsequent meta-analysis with the imputation of missing studies showed that podoplanin overexpression preserved the statistically significant results for the association with an increased risk of OPMD malignant transformation. Therefore, although our meta-analysis confirms the presence of publication bias, the results should be considered reliable and robust. In addition, a particularly noteworthy strength of this systematic review and meta-analysis is that all included studies were designed as longitudinal cohorts with patient follow-up over time. This design provides a higher level of evidence when compared to the predominantly cross-sectional studies typically used in the evaluation of other biomarkers in OPMDs. Consequently, this meta-analysis allows for a more precise estimation of causal associations and risk assessments, supporting more realistic and clinically relevant conclusions regarding the relationship between podoplanin overexpression and malignant transformation risk in OPMDs.

## 5. Conclusions

In conclusion, our evidence-based systematic review and meta-analysis demonstrates that podoplanin overexpression behaves as a biomarker of malignant transformation risk in the epithelium of oral potentially malignant disorders, singularly in oral leukoplakia. We believe that the routine application of immunohistochemistry in pathology laboratories makes it advisable to include podoplanin detection in the prognostic assessment of this oral potentially malignant disorder. The predictive value of podoplanin upregulation in other potentially malignant oral disorders, such as erythroplakia or oral lichen planus, also appears to be a promising tool, but further research is required.

## Figures and Tables

**Figure 1 cancers-17-03448-f001:**
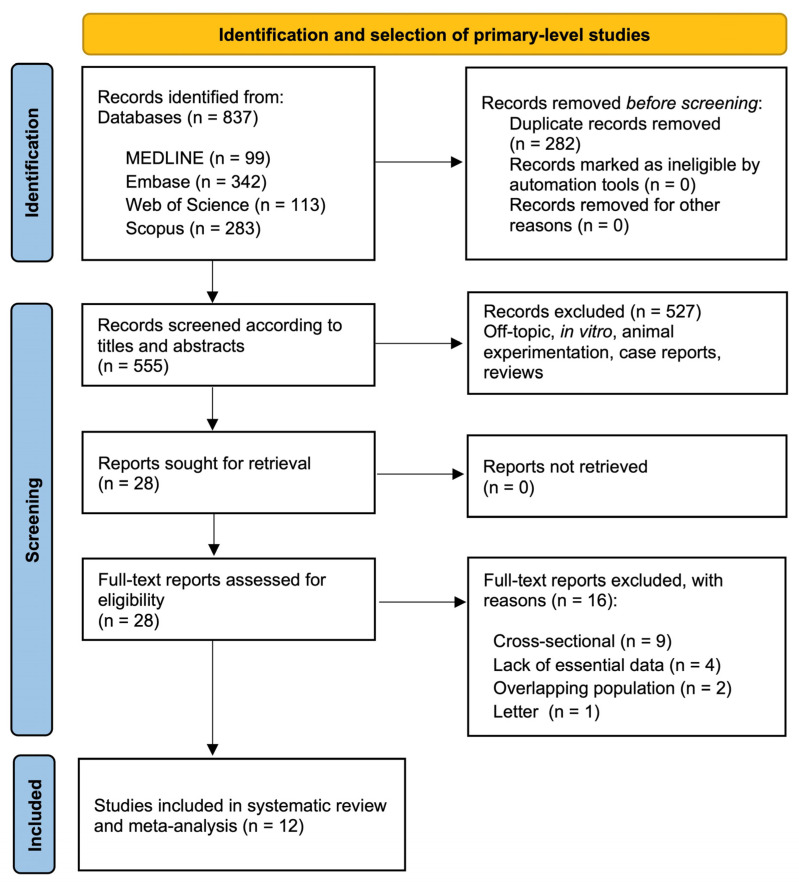
Flow diagram of the process of identification and selection of primary-level studies offering scientific information on the implications of podoplanin overexpression and the malignant transformation risk of OPMDs.

**Figure 2 cancers-17-03448-f002:**
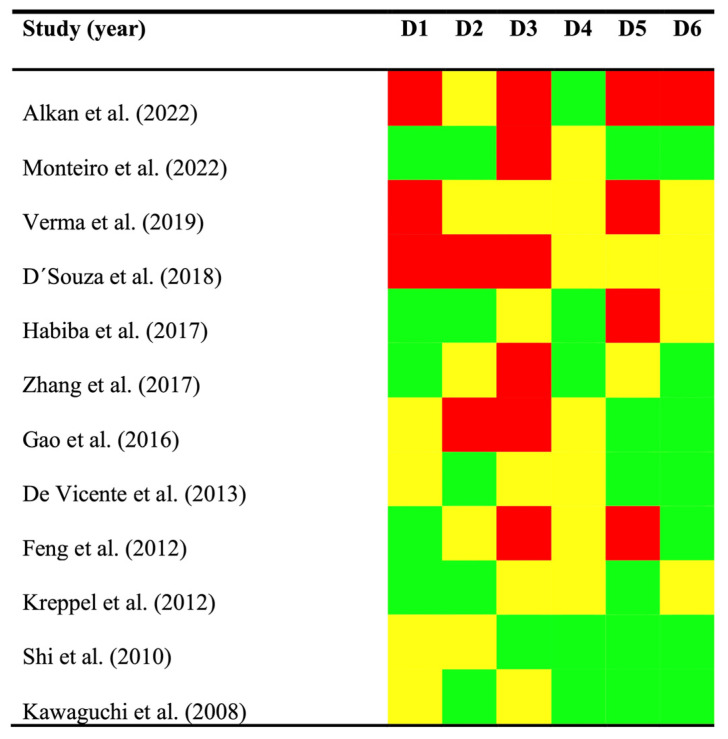
Quality plot graphically depicting the analysis of risk of bias (RoB) and methodological quality, critically judged by applying the QUIPS tool (developed by the Cochrane Prognosis Methods Group). The following six domains (D1–D6) were critically evaluated: *D1—Study participation*, *D2—Study attrition*, *D3—Prognostic factor measurement*, *D4—Outcome measurement*, *D5—Study confounding*, and *D6—Statistical analysis and reporting*. RoB was assessed for each primary study across all domains using a three-level scale reflecting a low, moderate, or high risk of bias. In the figure, studies with low potential bias are represented in green, those with moderate potential bias in yellow, and those with high potential bias in red.

**Figure 3 cancers-17-03448-f003:**
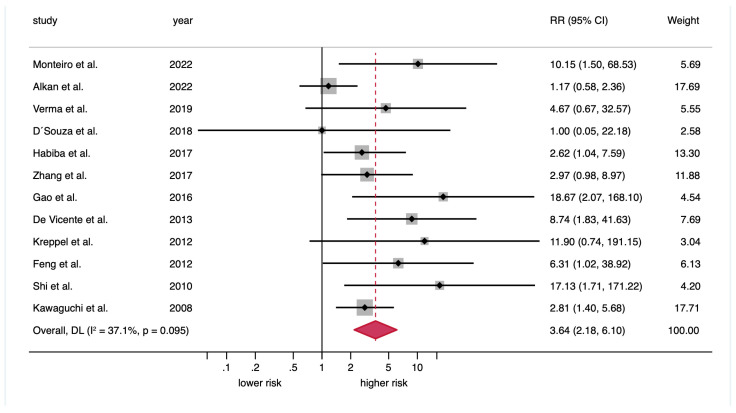
Forest plot graphically representing the meta-analysis of the association between podoplanin overexpression and OPMDs’ malignant transformation risk. RR, relative risk; CI, confidence interval; DL, DerSimonian and Laird. Random-effects model, inverse variance weighting based on the DL method. An RR > 1 suggests that podoplanin overexpression is associated with a higher malignant transformation risk. Diamonds indicate the pooled RR with their corresponding 95% CIs.

**Table 1 cancers-17-03448-t001:** Summarized characteristics of the study sample.

Total	12 studies
Year of publication	2008–2022
Total patients (range)	857 (30–160)
Type of OPMDs	
Oral leukoplakia	7 studies
Erythroplakia	1 study
Discoid lupus erythematosus	1 study
Oral lichen planus	1 study
Mixed	2 studies
Study design	
Retrospective	10 studies
Prospective	2 studies
Anti-podoplanin antibody	
Clone D2-40	10 studies
Not reported	2 studies
Anti-podoplanin antibody dilution	
1:100	6 studies
1:150	4 studies
Not reported	2 studies
Anti-podoplanin antibody incubation time	
Overnight	5 studies
60′	1 study
Not reported	6 studies
Anti-podoplanin antibody incubation temperature	
4 °C	5 studies
Room temperature	1 study
Not reported	6 studies
Cut-off point	
1%	4 studies
Intensity	2 studies
Not reported	6 studies
Immunostaining pattern	
Membrane and cytoplasm	6 studies
Membrane	4 studies
Cytoplasm	1 study
Not reported	1 study
Geographical region	
Asia	8 studies
Europe	3 studies
North America	1 study

**Table 2 cancers-17-03448-t002:** Meta-analyses of the predictive value of podoplanin overexpression on the malignant transformation risk of OPMD.

Meta-Analyses	No. of Studies	No. of Patients	Stat. Model	Wt	Pooled Data		Heterogeneity	
					ES (95% CI)	*p*-Value	*P_het_*	*I*^2^ (%)
Malignant transformation risk ^a^	12	857	REM	D-L	RR = 3.64 (2.18–6.10)	<0.001	0.10	37.1
Subgroup analysis by geographical region ^b^								
Asia	8	568	REM	D-L	RR = 3.20 (1.61–6.34)	0.001	0.09	42.6
Europe	3	157	REM	D-L	RR = 9.66 (3.19–29.26)	<0.001	0.98	0.0
North America	1	132	REM	D-L	RR = 2.81 (1.40–5.66)	0.004	—	0.0
Subgroup analysis by type of OPMD ^b^								
Oral leukoplakia	7	558	REM	D-L	RR = 3.37 (2.13–5.32)	<0.001	0.59	0.0
Erythroplakia	1	34	REM	D-L	RR = 6.31 (1.02–38.98)	0.05	—	0.0
Discoid lupus erythematosus	1	52	REM	D-L	RR = 18.67 (2.07–168.2)	0.009	—	0.0
Oral lichen planus	1	119	REM	D-L	RR = 17.13 (1.71–171.4)	0.02	—	0.0
Mixed	2	94	REM	D-L	RR = 1.72 (0.51–5.78)	0.38	0.19	42.1
Subgroup analysis by immunohistochemical pattern ^b^								
Membrane and cytoplasm	6	502	REM	D-L	RR = 4.71 (2.54–8.74)	<0.001	0.37	7.5
Membrane	4	291	REM	D-L	RR = 3.60 (1.96–6.62)	<0.001	0.50	0.0
Cytoplasm	1	30	REM	D-L	RR = 1.00 (0.05–21.06)	0.99	—	0.0
Not reported	1	34	REM	D-L	RR = 1.17 (0.58–2.36)	0.66	—	0.0
Subgroup analysis by anti-podoplanin antibody ^b^								
Clone D2-40	10	645	REM	D-L	RR = 3.45 (1.95–6.10)	<0.001	0.10	38.4
Not reported	2	212	REM	D-L	RR = 5.77 (1.02–32.55)	0.05	0.14	53.3
Subgroup analysis by anti-podoplanin antibody dilution ^b^						0.05		
1:100	6	523	REM	D-L	RR = 2.62 (1.51–4.56)	0.001	0.14	39.7
1:150	4	244	REM	D-L	RR = 11.04 (4.00–30.45)	<0.001	0.87	0.0
Not reported	2	90	REM	D-L	RR = 2.99 (0.58–15.39)	0.19	0.40	0.0
Subgroup analysis by anti-podoplanin antibody incubation time ^b^								
Overnight	5	448	REM	D-L	RR = 3.82 (2.11–6.92)	<0.001	0.33	12.8
60′	1	60	REM	D-L	RR = 4.67 (0.67–32.56)	0.12	—	0.0
Not reported	6	349	REM	D-L	RR = 3.46 (1.36–8.78)	0.009	0.05	54.7
Subgroup analysis by anti-podoplanin antibody incubation temperature ^b^								
4 °C	5	448	REM	D-L	RR = 3.82 (2.11–6.92)	<0.001	0.33	12.8
Room temperature	1	60	REM	D-L	RR = 4.67 (0.67–32.56)	0.12	—	0.0
Not reported	6	349	REM	D-L	RR = 3.46 (1.36–8.78)	0.009	0.05	54.7
Subgroup analysis by cut-off point for podoplanin protein overexpression ^b^								
1%	4	418	REM	D-L	RR = 3.42 (1.77–6.63)	<0.001	0.51	0.0
Intensity	2	73	REM	D-L	RR = 2.85 (0.35–22.90)	0.33	0.04	76.9
Not reported	6	366	REM	D-L	RR = 4.41 (2.32–8.38)	<0.001	0.36	8.3
Subgroup analysis by overall risk of bias in primary-level studies *^,b^								
Low RoB	4	369	REM	D-L	RR = 5.20 (2.16–12.51)	<0.001	0.26	26.1
Moderate RoB	6	420	REM	D-L	RR = 3.70 (2.01–6.82)	<0.001	0.48	0.0
High RoB	2	68	REM	D-L	RR = 2.18 (0.44–10.77)	0.34	0.09	65.1

Abbreviations: Stat., statistical; Wt, method of weighting; ES, effect size estimation; RR, relative risk; CI, confidence interval; REM, random-effects model; D-L, DerSimonian and Laird method; OPMD, oral potentially malignant disorders; RoB, risk of bias. * Note: RoB categories were defined according to an overall score derived from two critical domains (i.e., D3—prognostic factor measurement and D5—study confounding), identified as the main sources of bias in prognostic factor studies and following previously established methodology. Studies were classified as low, moderate, or high RoB based on their overall assessment across these domains. ^a^ Meta-analysis of aggregate (summary) data. ^b^ Subgroup meta-analyses.

## Data Availability

Data are contained within the article and Appendix A.

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
