# Peer review of "Implications of Podoplanin Overexpression in the Malignant Transformation of Oral Potentially Malignant Disorders: A Systematic Review and Meta-Analysis"

_cancers, 2025, doi:10.3390/cancers17213448_

Round 1

Reviewer 1 Report

Comments and Suggestions for Authors

Dear Authors,

The manuscript: "Implications of podoplanin overexpression in the malignant transformation of oral potentially malignant disorders: a systematic review and meta-analysis" comprises an interesting analysis of podoplanin overexpression in OPMDs. 

Here are my recommendations:

This statement  "Despite the importance of this topic, it is noteworthy that to date there is no evidence, in the form of systematic reviews and meta-analyses, on the implications of podoplanin in the malignant transformation of OPMDs" from the Introduction chapter as in August 2025, was published this systematic review https://pubmed.ncbi.nlm.nih.gov/40857023/

You can add the differences between your study and the previously mentioned one.

The Discussion chapter needs improvement to emphasize the connection of podoplanin with other forms of malignancies.

Best regards!

Comments on the Quality of English Language

There are some letters missing all over the manuscript.

Author Response

Reviewer 1

Comment 1

Dear Authors, The manuscript: "Implications of podoplanin overexpression in the malignant transformation of oral potentially malignant disorders: a systematic review and meta-analysis" comprises an interesting analysis of podoplanin overexpression in OPMDs. Here are my recommendations.

Response 1

We are very grateful for the insightful comments received from Reviewer 1. All were fully taken into account, addressed and replied in this new revised version of the manuscript.

Comment 2

This statement  "Despite the importance of this topic, it is noteworthy that to date there is no evidence, in the form of systematic reviews and meta-analyses, on the implications of podoplanin in the malignant transformation of OPMDs" from the Introduction chapter as in August 2025, was published this systematic review https://pubmed.ncbi.nlm.nih.gov/40857023/

You can add the differences between your study and the previously mentioned one.

Response 2

Thank you for this fine comment. We have revised the manuscript to include and discuss the recently published systematic review, which we were not aware of at the time of submission. Its inclusion enriches our work by allowing a more comprehensive comparison of the available evidence. Accordingly, the following paragraph has been added to the Discussion section:

“A systematic review and meta-analysis published during the final preparation of the present manuscript also examined the association between podoplanin expression and the risk of malignant transformation in OPMDs [11]. Their pooled sample partially overlapped with ours, and the meta-analytic findings were in line with our results, confirming that podoplanin overexpression was significantly associated with an increased risk of oral cancer development, with progressive higher podoplanin expression levels across the disease spectrum (i.e., normal oral mucosa < OPMD < OSCC) [11]. The study was well designed and conducted, however, one of the key aspects of the shared objective -i.e.,  to evaluate malignant transformation according to specific OPMD subtypes- was not addressed in their analysis [11]. In this regard, the present study represents the first systematic review and meta-analysis specifically designed to explore this dimension, including specific stratified meta-analytic results for oral leukoplakia, oral lichen planus, and oral erythroplakia, providing original and scientifically valuable insights that justify its publication.”.

In addition, the Introduction section has also been revised and the sentence stating the absence of previous systematic reviews and meta-analyses has now been removed for accuracy.

Comment 3

The Discussion chapter needs improvement to emphasize the connection of podoplanin with other forms of malignancies. Best regards!

Response 3

Following reviewer’s instructions, the Discussion section has been expanded to better emphasize the connection between podoplanin and other malignancies. We now discuss the association of podoplanin overexpression with poor prognosis and metastatic potential in oral, oropharyngeal, lung, esophageal, and ovarian cancers, highlighting its role in epithelial–mesenchymal transition and the acquisition of the hallmark invasion and metastasis, a key determinant of tumor progression and cancer-related mortality:

Podoplanin overexpression has also been associated with poorer prognosis in oral and oropharyngeal squamous cell carcinomas, including reduced overall survival, disease-specific survival, and disease-free survival, as well as a higher frequency of nodal metastasis (N+ status) [44]. Similar associations have been reported in other malignancies, most notably in lung, esophageal, and ovarian cancers [45,46] , where podoplanin expression correlates with unfavorable clinical outcomes. This behavior appears to be linked to the regulation of epithelial-mesenchymal transition (EMT) mediated by podoplanin [47,48], which confers tumor cells with a pro-migratory phenotype and provides an oncogenic advantage through the acquisition of the cancer hallmark “activating invasion and metastasis” [8,9]. This hallmark of cancer represents a critical step in tumor progression and remains one of the main determinants of cancer-related mortalityc [8,9,47,48]”

Reviewer 2 Report

Comments and Suggestions for Authors

Dear authors,

This is a full and complete review and met-analysis on podoplanin overexpression in the transformation of potential malignant disorders of the oral cavity.

This is a robust paper, that can add to the literature a comprehensive view of this interesting topic, with focus on methodological parameters associated with an high predictive consistency.

The abstract is clear.

The description of the background in the introduction is well-defined.

Data are presented quite clearly, with good methodical approach.

Conclusions are consistent with the evidence and arguments presented and clinically relevant.

No ethical problems were found.

No inflammatory material is found

Figures and tables are valid and clear

References are appropriate for the type of the study herein reported.

Suggestions:

I suggest to revise the paper for occasional grammatical and typographical errors

I suggest to, if it is possible, briefly expand the clinical implications of testing podoplanin in routinary practice

Author Response

Reviewer 2

Comment 1

Dear authors, This is a full and complete review and met-analysis on podoplanin overexpression in the transformation of potential malignant disorders of the oral cavity.

This is a robust paper, that can add to the literature a comprehensive view of this interesting topic, with focus on methodological parameters associated with an high predictive consistency.

The abstract is clear.

The description of the background in the introduction is well-defined.

Data are presented quite clearly, with good methodical approach.

Conclusions are consistent with the evidence and arguments presented and clinically relevant.

No ethical problems were found.

No inflammatory material is found

Figures and tables are valid and clear

References are appropriate for the type of the study herein reported.

Response 1

We are very grateful to Reviewer 2 for the thoughtful observations provided. We have carefully considered all suggestions and original ideas, which have been fully incorporated into and reflected in the revised version of the manuscript.

Comment 2

Suggestions:

I suggest to revise the paper for occasional grammatical and typographical errors. There are some letters missing all over the manuscript.

Response 2

Following the reviewer’s advice, the language and formatting have been extensively revised and improved throughout the entire text. All changes were highlighted in red in the revised version of the manuscript.

Comment 3

I suggest to, if it is possible, briefly expand the clinical implications of testing podoplanin in routinary practice

Response 3

Thank you again for this important comment. Following the reviewer’s instructions, we have now expanded the Discussion to highlight the potential clinical implications of assessing podoplanin overexpression in routine practice:

“Taken together, these findings support the incorporation of immunohistochemical assessment of podoplanin overexpression into routine clinical practice for patients with OPMDs, following the recommendations derived from this systematic review and meta-analysis, including the choice of anti-podoplanin clone, the immunostaining pattern, and the antibody’s incubation time and temperature. Immunohistochemistry is a simple, inexpensive, and widely available technique in pathology laboratories worldwide, and podoplanin could serve as a complementary prognostic tool, facilitating risk stratification for malignant transformation and potentially aiding in the early detection of oral cancer.”.

Reviewer 3 Report

Comments and Suggestions for Authors

The authors conducted a systematic review and meta-analysis to evaluate the potential predictive role of podoplanin overexpression in the malignant transformation of oral potentially malignant disorders. While the manuscript is interesting, several points should be addressed before it can be considered for publication:

1. Figure 1: Please provide details regarding the criteria for “Records excluded (n = 527)”.

2. Figure 2: The six domains (D1–D6) should be clearly defined in the figure legend. In addition, please specify the meanings of the red, yellow, and green colors to improve reader comprehension.

3. Tables 1 and 2: Please clarify the rationale for using 1% as the cut-off point, as well as “Intensity.” Were there previous publications that adopted this classification?

The subgroup label “Membrane/cytoplasm” should be revised to “Membrane & cytoplasm” to avoid potential confusion.

4. Table 2: Please define the criteria for low, moderate, and high risk of bias (RoB) in the table legend.

Author Response

Reviewer 3

Comment 1

The authors conducted a systematic review and meta-analysis to evaluate the potential predictive role of podoplanin overexpression in the malignant transformation of oral potentially malignant disorders. While the manuscript is interesting, several points should be addressed before it can be considered for publication.

Response 1

We are also very grateful to Reviewer 3 for the constructive and thoughtful comments, which have been carefully considered and incorporated into the revised manuscript. Your feedback has helped us improve the clarity and scientific value of the paper, and we truly appreciate your contribution to this process.

Comment 2

Figure 1: Please provide details regarding the criteria for “Records excluded (n = 527)”.

Response 2

In accordance with the reviewer’s suggestion, we have now provided the reasons for exclusion during the initial screening, identification, and selection phase based on titles and abstracts in the corresponding box of Figure 1 (i.e., off-topic, in vitro studies, animal experimentation, case reports, and reviews). We sincerely thank the reviewer for this valuable comment, which has enhanced the transparency and strengthened the systematic nature of the present study.

Comment 3

Figure 2: The six domains (D1–D6) should be clearly defined in the figure legend. In addition, please specify the meanings of the red, yellow, and green colors to improve reader comprehension.

Response 3

The legend of Figure 2 has been corrected according to the reviewer’s comment:

Figure 2. Quality plot graphically depicting the analysis of risk of bias (RoB) and methodological quality, critically judged by applying the QUIPS tool (developed by Cochrane Prognosis Methods Group). The following six domains (D1-D6) were critically evaluated:  D1-Study participation, D2-Study attrition, D3-Prognostic factor measurement, D4-Outcome measurement, D5-Study confounding, and D6-Statistical analysis and reporting. RoB was assessed for each primary study across all domains using a three-level scale reflecting low, moderate, or high risk of bias. In the figure, studies with low potential bias are represented in green, those with moderate potential bias in yellow, and those with high potential bias in red.”

Comment 4

Tables 1 and 2: Please clarify the rationale for using 1% as the cut-off point, as well as “Intensity.” Were there previous publications that adopted this classification?

Response 4

Thank you again for this comment. Following the reviewer’s instructions, we have now clarified this important aspect in the Discussion section:

“Most research groups quantified podoplanin overexpression using a 1% cut-off point or minor adjustments, as podoplanin expression in the oral epithelium is considered an abnormal feature [61]. The most commonly used cut-off defined a sample as positive when podoplanin expression was observed in at least one area of the suprabasal layer (also defined as >1% suprabasal expression), with or without basal expression, based on the rationale that this may reflect clonal expansion and a higher risk of malignant progression [40]. Regarding studies that used staining intensity as a criterion, the methodology involved scoring samples according to staining intensity alone (‘none’, ‘weak’, ‘moderate’, or ‘strong’) [32], or in combination with the percentage of stained cells [42].”

Comment 5

The subgroup label “Membrane/cytoplasm” should be revised to “Membrane & cytoplasm” to avoid potential confusion.

Response 5

The term “Membrane/cytoplasm” has been revised and is now referred to as “Membrane & cytoplasm” in the subgroup label of Table 2, as well as throughout the whole manuscript.

Comment 6

Table 2: Please define the criteria for low, moderate, and high risk of bias (RoB) in the table legend.

Response 6

Following the reviewer’s instruction the legend of Table 2 has been corrected as follows:

“*-Note: RoB categories were defined according to an overall score derived from two critical domains (i.e., D3-Prognostic factor measurement and D5-study confounding), identified as the main sources of bias in prognostic factor studies and following previously established methodology. Studies were classified as low, moderate, or high RoB based on their overall assessment across these domains.”

We sincerely thank again all reviewers for their critical comments, which has undoubtedly helped us to substantially improve the quality, clarity, and scientific rigor of the present manuscript. We have carefully addressed all comments and revised the text accordingly. We truly hope that the revised version now meets the standards for publication and can be favorably considered.